

# Microbiome dynamics in the tissue and mucus of acroporid corals differ in relation to host and environmental parameters

Giulia M. Marchioro[1,2,3,*], Bettina Glasl[3,4,5,*], Aschwin H. Engelen[2], Ester A. Serrão[2], David G. Bourne[3,4,5], Nicole S. Webster[3,4,6] and Pedro R. Frade[2]

[1] University of Algarve, Faro, Portugal
[2] CCMAR - Centre of Marine Sciences, University of Algarve, Faro, Portugal
[3] AIMS@JCU, Townsville, Queensland, Australia
[4] Australian Institute of Marine Science, Townsville, Queensland, Australia
[5] College of Science and Engineering, James Cook University, Queensland, Townsville, Australia
[6] Australian Centre for Ecogenomics, University of Queensland, Brisbane, Queensland, Australia
[*] These authors contributed equally to this work.

Corresponding author
Bettina Glasl, b.glasl@aims.gov.au

## ABSTRACT

Corals are associated with diverse microbial assemblages; however, the spatial-temporal dynamics of intra-species microbial interactions are poorly understood. The coral-associated microbial community varies substantially between tissue and mucus microhabitats; however, the factors controlling the occurrence, abundance, and distribution of microbial taxa over time have rarely been explored for different coral compartments simultaneously. Here, we test (1) differentiation in microbiome diversity and composition between coral compartments (surface mucus and tissue) of two *Acropora* hosts (*A. tenuis* and *A. millepora*) common along inshore reefs of the Great Barrier Reef, as well as (2) the potential linkage between shifts in individual coral microbiome families and underlying host and environmental parameters. Amplicon based 16S ribosomal RNA gene sequencing of 136 samples collected over 14 months, revealed significant differences in bacterial richness, diversity and community structure among mucus, tissue and the surrounding seawater. Seawater samples were dominated by members of the Synechococcaceae and Pelagibacteraceae bacterial families. The mucus microbiome of *Acropora* spp. was dominated by members of Flavobacteriaceae, Synechococcaceae and Rhodobacteraceae and the tissue was dominated by Endozoicimonaceae. Mucus microbiome in both *Acropora* species was primarily correlated with seawater parameters including levels of chlorophyll *a,* ammonium, particulate organic carbon and the sum of nitrate and nitrite. In contrast, the correlation of the tissue microbiome to the measured environmental (i.e., seawater parameters) and host health physiological factors differed between host species, suggesting host-specific modulation of the tissue-associated microbiome to intrinsic and extrinsic factors. Furthermore, the correlation between individual coral microbiome members and environmental factors provides novel insights into coral microbiome-by-environment dynamics and hence has potential implications for current reef restoration and management efforts (e.g. microbial monitoring and observatory programs).

## INTRODUCTION

Coral microbiomes include the well-characterized endosymbiotic dinoflagellates of the family Symbiodiniaceae, and a vast diversity of bacteria and archaea (*Bourne, Morrow & Webster, 2016*; *Frade et al., 2016a*; *Rohwer et al., 2002*). The microbiome has a fundamental role in the health and stability of the coral holobiont; it recycles nutrients, removes waste products and defends against pathogens (*Lema, Willis & Bourne, 2012*; *Morris et al., 2011*; *Rädecker et al., 2015*; *Rosado et al., 2019*). The coral microbiome is influenced by a variety of intrinsic and extrinsic factors. Coral microbiomes are host species-specific and were thought to remain relatively stable over space and time (*Frias-Lopez et al., 2002*; *Rohwer et al., 2002*). However, recent studies have proposed that spatial–temporal factors such as environmental parameters (*Chen et al., 2011*), depth (*Glasl et al., 2017*), geography (*Hong et al., 2009*; *Littman et al., 2009*), seasonality (*Ceh, Van Keulen & Bourne, 2011*; *Chen et al., 2011*; *Hong et al., 2009*; *Koren & Rosenberg, 2006*), coastal pollution (*Klaus et al., 2007*), and the physiological status of the host (*Grottoli et al., 2018*; *Littman, Willis & Bourne, 2009*) can also influence the occurrence and relative abundance of microbial taxa. For instance, *Li et al. (2015)* reported a dynamic relationship between the community structure of coral-associated bacteria and the seasonal variation in environmental parameters such as dissolved oxygen and rainfall. *Glasl et al. (2019a)* showed that although host-associated microbiomes were five-times less responsive to the environment compared to the seawater microbiome, they were still affected by environmental factors (e.g., temperature, turbidity, and nutrient concentration).

The coral provides different microhabitats for its microbial associates, including the surface mucus layer, coral tissue, skeleton and gastrovascular cavity, each differing in microbial richness, diversity and community structure, often assessed through alpha- and beta-diversity metrics (*Agostini et al., 2012*; *Engelen et al., 2018*; *Pollock et al., 2018*; *Sweet, Croquer & Bythell, 2011*). Each microhabitat has a unique set of biochemical features and harbors a specific microbial community (*Engelen et al., 2018*; *Pollock et al., 2018*; *Sweet, Croquer & Bythell, 2011*). Hence, revealing microhabitat-specific host-microbiome associations and their specific sensitivities to environmental fluctuations is crucial to our understanding of coral holobionts. For example, the coral surface mucus layer is a polysaccharide-protein-lipid complex that provides an interface between the coral epithelium and the surrounding seawater (*Brown & Bythell, 2005*). Here microbes take advantage of a nutrient-rich medium and particular microbiome members found in the coral mucus overlap with both the tissue and the seawater microbial communities (*Bourne & Munn, 2005*; *Brown & Bythell, 2005*; *Glasl, Herndl & Frade, 2016*; *Sweet, Croquer & Bythell, 2011*). In contrast to the extracellular polymeric nature of the surface mucus layer, the coral tissue consists of two distinct layers (epidermis and gastrodermis) and a connective-tissue layer, the mesoglea (*Muller-Parker, D'Elia & Cook, 2015*). The coral

tissue harbors photosymbiotic dinoflagellates (family Symbiodiniaceae), that can provide up to 100% of energy required by their coral host (*Muller-Parker, D'Elia & Cook, 2015*). The Symbiodiniaceae community has been shown to vary in tandem with the bacterial community in early life stages of corals (*Quigley et al., 2019*) and this may be caused by the release of complex organic molecules such as the organosulfur compound dimethylsulfoniopropionate (DMSP; *Bourne et al., 2013*; *Frade et al., 2016b*). The coral tissue microbiome is mostly represented by bacteria belonging to the phyla Proteobacteria and Actinobacteria. For example, the gammaproteobacterial *Endozoicomonas* are abundant in the coral's endodermal tissue and are often considered 'true' coral symbionts (*Bayer et al., 2013*; *Glasl et al., 2019b*; *Neave et al., 2016*; *Neave et al., 2017*). When compared to the surface mucus layer, the microbial community in the tissue is significantly less dense and diverse (*Bourne & Munn, 2005*; *Koren & Rosenberg, 2006*), likely attributed to the more spatially stable and host controlled environment (*Bourne & Munn, 2005*), although divergent evidence suggests the mucus is less diverse than the tissue (*Pollock et al., 2018*). Furthermore, tissue-associated bacterial communities form aggregations within the coral cell layers, also referred to as coral-associated microbial aggregates (CAMAs), and are often co-localized near algal symbiont cells, highlighting potential metabolic interactions between symbionts (*Wada et al., 2019*).

In this study, we test the hypotheses that different coral compartments (surface mucus layer and tissue) of *Acropora* spp. harbor distinct microbial communities and that different intrinsic and extrinsic factors explain microbiome dynamics within these compartments. Furthermore, we aim to identify significant correlations of individual bacterial families associated with coral tissue and mucus with host-physiological and seawater parameters.

## MATERIALS & METHODS

### Sample collection

Samples of *Acropora millepora*, *Acropora tenuis* and seawater were collected monthly, at Geoffrey Bay (Magnetic Island) in the Great Barrier Reef (Fig. S1), between February 2016 and March 2017, for amplicon based 16S ribosomal RNA (rRNA) gene sequencing along with environmental metadata. All samples were collected under the permit G16/38348.1 issued by the Great Barrier Reef Marine Park Authority.

Samples ($n = 3$ per sample type and per sampling event) for molecular analysis were collected as part of the Australian Microbiome Initiative and the sample procedure has previously been outlined by *Glasl et al. (2019a)*. In brief, coral nubbins (approximately 5 cm tall) of both *Acropora* species were collected, rinsed with 0.2 μm filter-sterilized seawater and placed into cryogenic vials. Coral mucus from the same specimens was collected with sterile cotton swabs as previously described by *Glasl, Herndl & Frade (2016)*. Seawater samples for molecular analysis were collected in sterile collapsible bags, pre-filtered through a 50 μm filter mesh to remove large particles, and subsequently filtered onto a 0.2 μm Sterivex filter (Millipore). Coral nubbins, mucus swabs and Sterivex filters were immediately snap frozen in liquid nitrogen after collection and stored at −80 °C until further processing. To acquire environmental information, water and sediment samples were collected in
duplicate for each sampling event as described in *Glasl et al. (2019a)* and further analyzed according to the standard procedures of the Australian Institute of Marine Science (AIMS; *Devlin & Lourey, 2000*). The environmental information processed includes common reef water quality measures such as salinity, particulate organic carbon, total suspended solids, concentrations of chlorophyll *a,* ammonium, the sum of nitrite and nitrate, particulate nitrogen, nitrite, total nitrogen, non-purgeable organic carbon, non-purgeable inorganic carbon, phosphate and silica as well as total organic carbon in the sediment, total organic nitrogen in the sediment and grainsize percentage of sediments <0.63 $\mu$m, between 0.63 $\mu$m and 2 mm, and >2 mm. Seawater temperatures and daylight hours were obtained from AIMS long-term monitoring temperature records (http://eatlas.org.au).

## Sample preparation and genetic assays

Frozen coral tissue was airbrushed into a *ziploc* bag with phosphate-buffered saline (PBS) solution added until all tissue was removed from the skeletal fragment (total PBS volume was recorded). The resulting tissue slurry was homogenized for 1 min at 12,500 rpm using a hand-held tissue homogenizer (Heidolph Silent Crusher M), pelleted (10 min at 16,000 rcf) and snap frozen in liquid nitrogen. DNA from the tissue and mucus samples was extracted using the DNeasy PowerBiofilm kit (QIAGEN). DNA extracts were sent on dry ice to the Ramaciotti Centre for Genomics (Sydney, Australia) for sequencing. The bacterial 16S rRNA gene was sequenced using the 27F (*Lane, 1991*) and 519R (*Turner et al., 1999*) primers on the Illumina MiSeq platform using a dual indexed 2 × 300 bp paired-end approach. Primer pairs were selected to warrant comparability across datasets of the Australian Microbiome Initiative (https://www.australianmicrobiome.com).

## Sequence analysis

Sequencing data were analyzed as single nucleotide variants following the standardized platform of the Australian Microbiome Initiative (*Brown et al., 2018*). In brief, paired-end reads were merged using FLASH software (*Magoc & Salzberg, 2011*) and FASTA formatted sequences were extracted from FASTQ files. Sequences <400 bp in length, and / or containing one or more N's, or homopolymer runs of >8 bp were removed with MOTHUR (v1.34.1; *Schloss et al., 2009*). Sequences were de-replicated and ordered by abundance using USEARCH (64 bit v10.0.240; *Edgar, 2010*). Sequences with less than 4 representatives and Chimeras were removed, and the quality-filtered sequences were mapped to chimera-free zero-radius operational taxonomic units (zOTUs). A table containing the samples and their read abundances was created and the zOTUs were taxonomically classified with SILVA v132 database (*Yilmaz et al., 2014*) using MOTHUR's implementation of the Wang classifier (*Wang et al., 2007*) and a 60% Bayesian probability cut-off. This sequencing dataset has already been used in a previous contribution by the research group (*Glasl et al., 2019a*), but in the current study it is analyzed from a different perspective aiming at comparing temporal microbiome dynamics between two distinct coral compartments.

Chloroplasts and mitochondria derived reads were removed from the dataset and remaining data was rarefied to a sequencing depth of 3,500 reads per sample in R (*R Core Team, 2015*) using subset_taxa function in the phyloseq package (*McMurdie & Holmes, 2013*). Read counts per sample were transformed into relative abundances.

## Coral holobiont photopigment quantification

Photopigment (chlorophyll *a*) concentrations in the tissue of corals were quantified using a spectrophotometric approach (*Glasl et al., 2019b*). Tissue pellets were thawed on ice to avoid sample degradation and resuspended in 1 ml of 90% ethanol. Samples were sonicated for 1 min and centrifuged for 5 min at 10,000 rcf. Subsequently, 700 µl of the supernatant was removed and transferred to a new tube. The resuspension, sonication and centrifugation were repeated on the remainder of the pellet. The supernatant was recovered again, combined with the previous extraction and mixed by inversion. Sample extract and 90% ethanol (blank read) were loaded in triplicate (200 µl each) to a 96-well plate and the absorbance was recorded at 470, 632, 649, 665, 696 and 750 nm in a Cytation 3 multi-mode microplate reader (BioTek, Winooski, USA) and analyzed using the software Gen5 (BioTek, Winooski, USA). Blank corrected absorbance measures were used to calculate chlorophyll *a* concentrations (Equation S1).

## Coral protein quantification

Soluble protein concentrations of coral tissue samples were quantified using a colorimetric protein assay kit (Pierce BCA Protein Assay Kit; *Glasl et al., 2019b*). Tissue pellets were thawed on ice and resuspended in 1 ml PBS. The resuspension (25 µl) was added to 200 µl of working reagent from the kit in a 96-well plate. The plate was mixed thoroughly on a plate shaker for 30 s and then incubated at 37 °C for 30 min. The plate was cooled down at room temperature. The absorbance was measured at 563 nm in a Cytation 3 multi-mode microplate reader (BioTek, Winooski, USA) and analyzed using the software Gen5 (BioTek, Winooski, USA). Measurements of the standards and samples were blank corrected to remove background absorbance. For each plate, a protein standard curve was obtained using bovine serum albumin (BSA) solution at concentrations between 25 and 2,000 µg ml$^{-1}$.

## Symbiodiniaceae cell counting

To determine cell numbers of Symbiodiniaceae in the coral tissue, the tissue pellet was thawed on ice, resuspended in 1 ml of 0.2 µl filtered seawater and fixed in 2% formaldehyde (final concentration) to preserve the symbiont cells. The solution was passed through a syringe needle to reduce cell agglomeration and diminish the bias from cell clumps. Samples were then mixed for 1 min and 10 µl of the homogenate was loaded onto a Neubauer haemocytometer (0.100 mm depth). Symbiodiniaceae cells were counted under 40× magnification with an Olympus CX31 light microscope. In total, six independent haemocytometer loadings (24 squares each with 0.1 µl volume) were used per sample to ensure robustness of density determinations.

## Statistical analyses

Statistical analyses were performed using RStudio (v1.1.463). Analyses of microbial communities were performed on rarefied relative abundance data at zOTU level. zOTU richness and Shannon-Weaver diversity were compared across host compartments, host species and reference seawater samples using non-parametric Analysis of Variance (Kruskal-Wallis test using function kruskal.test), followed by Dunn's test for multiple

comparisons (function dunn.test). All *p*-values were adjusted using the Benjamini–Hochberg multiple comparison correction method to decrease the false discovery rate (*Benjamini & Hochberg, 1995*). A Venn diagram was constructed to describe the shared and unique zOTUs among mucus, tissue and seawater microbiomes using VennDiagram package (*Chen & Boutros, 2011*) and visualized using eulerr package (*Larsson, 2020*). Non-Metric Multidimensional Scaling (NMDS) was used to illustrate the microbial community structure among host species and host compartments based on Bray-Curtis dissimilarities (phyloseq package *McMurdie & Holmes, 2013*). Permutational Multivariate Analysis of Variance (PERMANOVA, 999 permutations) was used to test for differences in microbial structure between host species and host compartments using the adonis2 function of the vegan package (*Oksanen et al., 2013*).

Physiological variables were normalized (i.e., chlorophyll *a* normalized to protein content, chlorophyll *a* normalized to Symbiodiniaceae numbers, Symbiodiniaceae density normalized to protein content) following common procedures in coral physiology studies (*Frade et al., 2008*; *Iglesias-Prieto & Trench, 1997*). Due to fragmentation of the collected coral branches, coral surface area could not be measured. Environmental and physiological variables were standardized and checked for collinearity using the Pearson correlation coefficient. Redundant variables based on Pearson's correlation (>0.7 or <-0.7; *Dormann et al., 2013*) were removed from the analysis. Non-correlated variables were then used in a Bray-Curtis distance-based Redundancy Analysis (db-RDA), which quantifies the impact of the explanatory variables on the microbiome (dis)similarities (*Legendre & Anderson, 1999*). zOTU relative abundance, environmental and physiological metadata were used for db-RDA using the phyloseq package (*McMurdie & Holmes, 2013*). The analysis tests the statistical relationship between microbial community composition and the environmental/physiological variables for each coral compartment and host species combination. A model selection tool (ordiR2step function in the vegan package, *sensu Blanchet, Legendre & Borcard, 2008*) was performed to select the best db-RDA model (i.e., the best explanatory variables) for variation in microbiome composition of each coral compartment (mucus and tissue) in each host species (*Johnson & Omland, 2004*). The significance of each explanatory variable was confirmed with an ANOVA-like permutational test (function permutest) for dbRDA. The explanatory value (in %) of significant explanatory variables (e.g., environmental and physiological parameters) on each microbiome was assessed with Variation Partitioning Analysis of the vegan package (*Oksanen et al., 2013*). A correlation matrix (based on the default Pearson correlation) between the relative abundance of the 20 most abundant microbial families and significant environmental variables was generated using the R package MicrobiomSeq (*Ssekagiri, Sloan & Ijaz, 2017*), for which *p*-values were adjusted using the Benjamini–Hochberg multiple comparison correction (*Benjamini & Hochberg, 1995*).

# RESULTS

## Composition of coral tissue and mucus microbiomes

The bacterial 16S rRNA genes derived from 136 samples, including coral tissue ($n = 24$ for *A. millepora*; $n = 30$ for *A. tenuis*), coral mucus layer ($n = 24$ for *A. millepora*; $n = 28$

for *A. tenuis*) and seawater ($n = 30$; used as reference samples) were sequenced and 12,051 zOTUs identified as single nucleotide variants.

zOTU richness differed significantly among mucus, tissue and seawater microbiomes (Kruskal-Wallis $Chi^2_{(2,133)} = 57.74$, $p = 2.89 \times 10^{-13}$), but not between seasons (see Table S1). Coral zOTU richness differed between species (*A. millepora vs A. tenuis*; Kruskal-Wallis $Chi^2_{(1,134)} = 12.23$, $p = 0.00047$). Seawater harbored the richest microbial community (558 zOTU $\pm$ 54.6), followed by the mucus (*A. millepora*, 220 zOTU $\pm$ 188; *A. tenuis* 511 zOTU $\pm$ 234) and tissue (*A. millepora*, 125 zOTU $\pm$ 31.6; *A. tenuis*, 173 zOTU $\pm$ 146; Table S1). Alpha diversity based on Shannon Index also differed significantly among microbiomes from mucus, tissue and seawater (Kruskal-Wallis $Chi^2_{(2,133)} = 53.37$, $p = 2.57 \times 10^{-12}$), but not between seasons (see Table S1). Coral zOTU Shannon differed between species (*A. millepora vs A. tenuis*; Kruskal-Wallis $Chi^2_{(1,134)} = 6.002$, $p = 0.01429$). Alpha diversity measures of mucus samples were not significantly different (Shannon Index: *A. millepora*, 4.18 $\pm$ 0.83; *A. tenuis*, 5.15 $\pm$ 0.69) from seawater samples (Shannon Index: 4.40 $\pm$ 0.209; Table S1). In contrast, the tissue microbiome was dramatically different from the mucus and seawater microbiomes and harbored the lowest microbial diversity (Shannon Index: *A. millepora*, 3.35 $\pm$ 0.63; *A. tenuis*, 3.54 $\pm$ 0.84).

Sequences affiliated to the phyla Proteobacteria dominated the microbial community of all samples (average relative abundance $\pm$ SD; mucus: 44.1 $\pm$ 11.5%; tissue: 62.8 $\pm$ 2%; seawater: 39.6 $\pm$ 3.1%), followed in dominance by Bacteroidetes (mucus: 27.5 $\pm$ 13.0%; tissue: 9.6 $\pm$ 10.9%; seawater: 12.0 $\pm$ 11.4%) and Cyanobacteria (mucus: 14.4 $\pm$ 9.0%; tissue: 9.8 $\pm$ 11.0%; seawater: 38.5 $\pm$ 4.0%). Mucus microbiomes for both *Acropora* species (Fig. 1) were characterized mostly by members of the family Flavobacteriaceae (average relative abundance $\pm$ SD; for *A. tenuis*: 17.3 $\pm$ 9.1%; *A. millepora*: 17.3 $\pm$ 12.7%), Synechococcaceae (*A. tenuis*: 12.3 $\pm$ 7.8%; *A. millepora*: 13.1 $\pm$ 10.2%) and Rhodobacteraceae (*A. tenuis*: 5.7 $\pm$ 3.0%; *A. millepora*: 6.4 $\pm$ 6.4%; Fig. 1). In contrast, the Endozoicimonaceae family dominated the tissue microbiome (*A. tenuis*: 43.2 $\pm$ 31.7%; *A. millepora*: 20.5 $\pm$ 19.7%), with additional representation of Flavobacteriaceae (*A. tenuis*: 7.9 $\pm$ 9.6%; *A. millepora*: 7.2 $\pm$ 9.6%), Synechococcaceae (*A. tenuis*: 5.5 $\pm$ 6.8%; *A. millepora*: 12.3 $\pm$ 14.5%) and Rhodobacteraceae (*A. tenuis*: 6.5 $\pm$ 10.4%; *A. millepora*: 5.3 $\pm$ 8.5%; Fig. 1) families. Seawater samples were mostly characterized by members of Synechococcaceae (36.6 $\pm$ 3.9%) and Pelagibacteraceae (18.6 $\pm$ 4.9%), but also by Rhodobacteraceae (8.6 $\pm$ 4.8%) and Flavobacteriaceae (8.0 $\pm$ 2.6%; Fig. 1). Tissue and mucus microbiomes exclusively shared 1,193 zOTUs (9.9%), mucus and seawater microbiomes exclusively shared 1,458 zOTUs (12.1%), whereas the tissue and seawater microbiome shared only 66 zOTUs (0.6%; Fig. 2).

Microbial community composition (beta-diversity) significantly differed among mucus, tissue and seawater (Fig. 3; PERMANOVA, *pseudo-F*$_{(2,126)} = 14.53$, $p = 0.001$), between *Acropora* species (PERMANOVA, *pseudo-F*$_{(1,126)} = 4.42$, $p = 0.001$), and between seasons (PERMANOVA, *pseudo-F*$_{(1,126)} = 1.90$, $p = 0.011$). Interaction between species and compartment was also significant (PERMANOVA, *pseudo-F*$_{(1,126)} = 3.07$, $p = 0.002$; other interactions were not significant; Table S2).

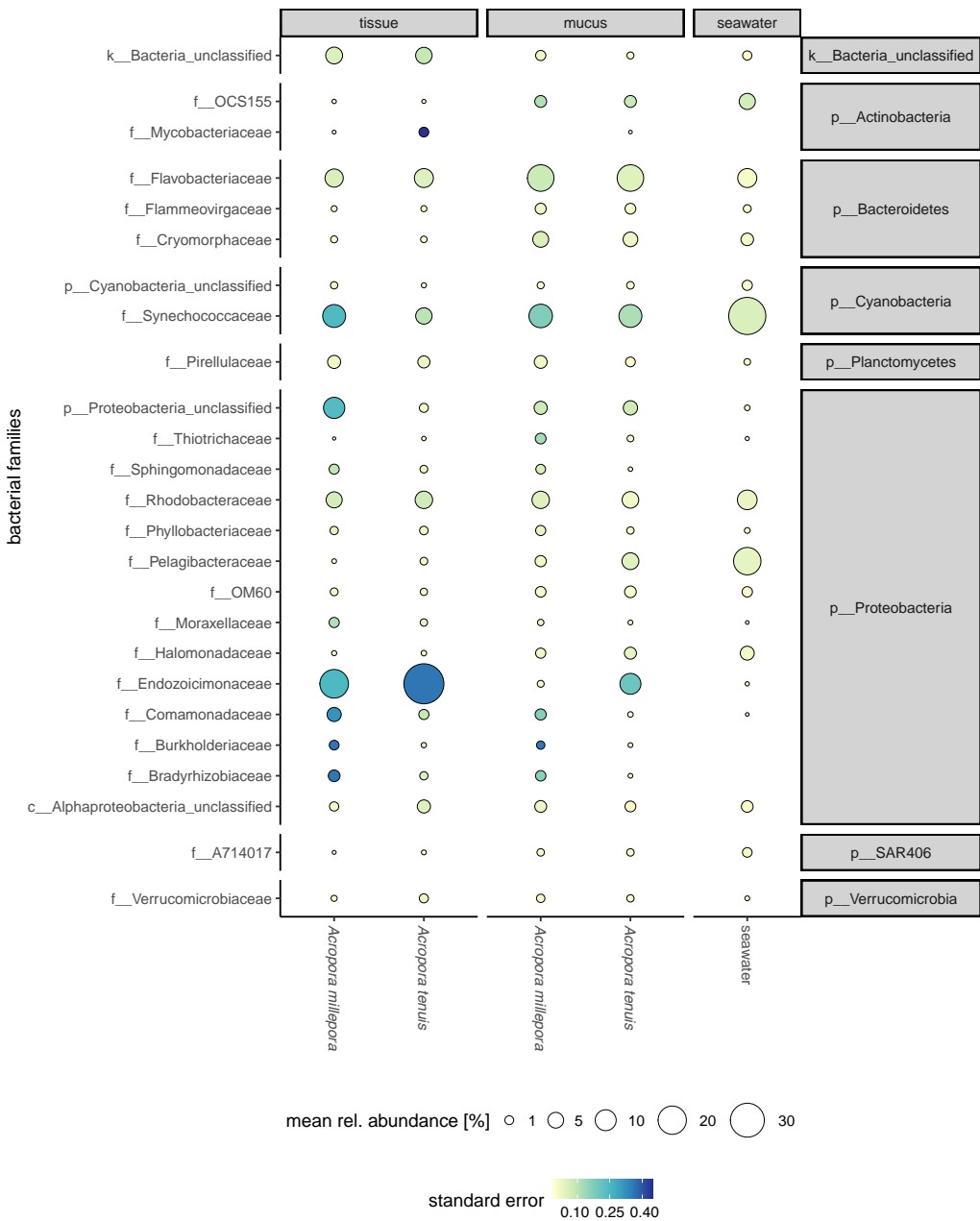

**Figure 1** **Compartment-specific microbiome composition of *Acropora tenuis* and *Acropora millepora*.** Microbial community composition (size indicates mean relative abundance and color represents standard error) resolved for the surface mucus layer and tissue of two *Acropora* coral species (*A. tenuis* and *A. millepora*), and surrounding seawater, based on partial 16S rRNA gene amplicon sequencing. Only the 25 most abundant families across all samples are represented.

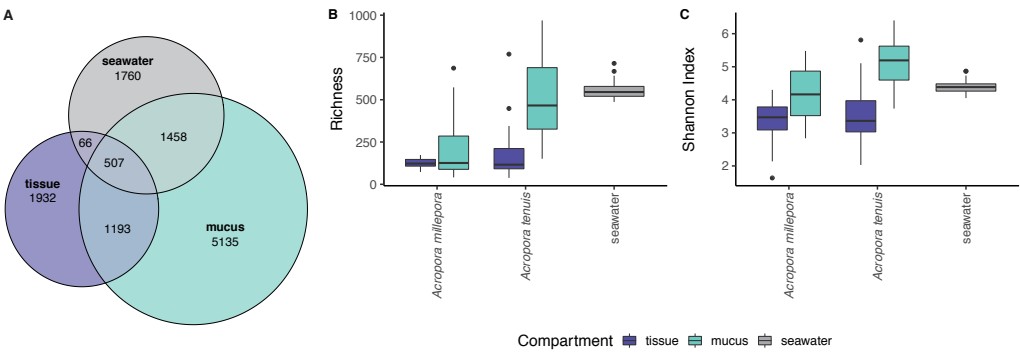

**Figure 2 Alpha diversity measures of coral mucus, coral tissue, and seawater microbiomes.** (A) Venn diagram displaying the number of shared, unique, and ubiquitous zOTUs among mucus, tissue and seawater microbiomes. Two *Acropora* species (*A. tenuis* and *A. millepora*) are pooled for the tissue and mucus microbiomes. (B) zOTU richness and (C) Shannon diversity index of microbiomes associated with tissue and mucus of *A. millepora* and *A. tenuis*, as well as with seawater.

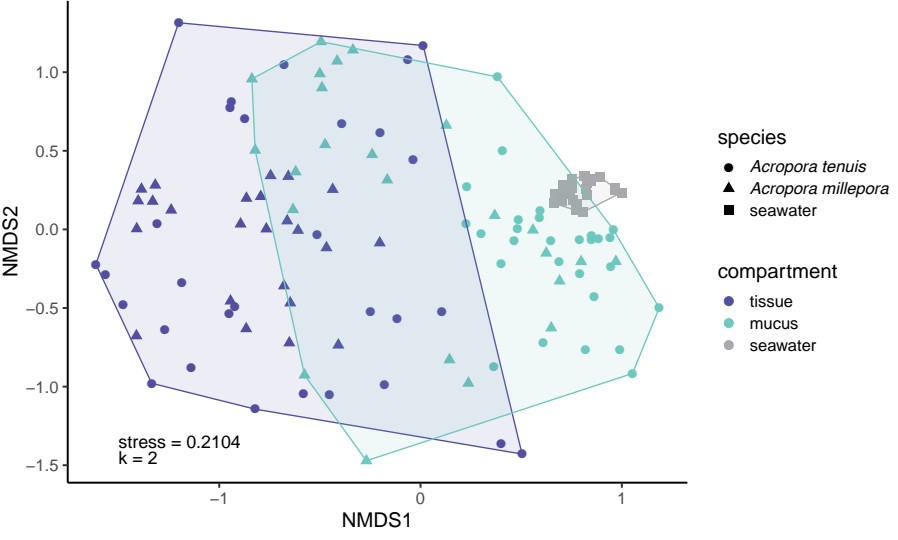

**Figure 3 Compositional variability of microbiomes associated with distinct coral-compartments and the ambient seawater.** Two-dimensional non-metric multidimensional scaling (nMDS) ordination depicting variation in microbial community structure between coral compartments (mucus and tissue) of *Acropora tenuis and Acropora millepora*, and seawater samples. "k" is the number of dimensions.

## Explanatory variables of coral tissue and mucus microbiomes

Physiological parameters of the tissue (i.e., chlorophyll *a* normalized to protein content, chlorophyll *a* normalized to Symbiodiniaceae numbers, Symbiodiniaceae density normalized to protein content) remained stable between host species.

Out of a total of 20 environmental variables measured for seawater and sediment, 6 variables were non-mutually collinear and were thus included in the db-RDA analysis. Selected variables were salinity, concentration of particulate organic carbon (POC), total

suspended solids (TSS), chlorophyll $a$ (Chl$a$), ammonium ($NH_4^+$) and the sum of nitrite and nitrate concentrations (i.e., $NO_2^-/NO_3^-$; Table S3).

Environmental/physiological parameters investigated in this study explained a limited amount of variation in the microbial community of mucus and tissue of the two *Acropora* species studied (Fig. 4). For example, seawater parameters explained 14% (Chl$a$, $NH_4^+$ and $NO_2^-/NO_3^-$) and 10% (POC and $NO_2^-/NO_3^-$) of the compositional variability for the mucus microbiome in *A. tenuis* and *A. millepora*, respectively (ANOVA-like permutational test for dbRDA; Table S4); $NO_2^-/NO_3^-$ was the only explanatory environmental variable common to the mucus microbiome of both *Acropora* species (5% of compositional variability explained in each species). In comparison, for the seawater microbiome, environmental parameters ($NO_2^-/NO_3^-$, TSS, POC, Salinity and Chl$a$) explained 32% of the compositional variability of the microbiome (Fig. S2), suggesting greater environmental sensitivity by the microbial community in the seawater compared to the coral-associated communities.

In contrast, tissue microbiomes of *A. millepora* and *A. tenuis* differed substantially in their response to environmental and/or to physiological parameters. While host physiology (i.e., Symbiodiniaceae density normalized to protein contents) and environment (TSS and Chl$a$) explained 6% and 10%, respectively, of the variation of the tissue microbiome in *A. tenuis*, in *A. millepora*, the compositional variation was solely explained (10%) by environmental parameters ($NO_2^-/NO_3^-$ and TSS; Variation Partitioning Analysis and ANOVA-like permutational test for dbRDA; Table S4). TSS was the only explanatory environmental variable common to the tissue microbiomes of both *Acropora* species (total of 5% and 4% in *A. tenuis* and in *A. millepora*, respectively).

## Correlation between bacterial families and environmental/physiological parameters

The relative abundance of Synechococcaceae derived from tissue samples of both *Acropora* species and the mucus of *A. tenuis* was negatively correlated with TSS ($p = 0.025 - 0.039$; Fig. 5 and Tables S5 and S6). In contrast, Synechococcaceae was positively correlated to total $NO_2^-/NO_3^-$ in both species (mucus of *A. tenuis*, $p = 0.002$, Table S5; and tissue of *A. millepora*, $p = 0.024$, Tables S5). For *A. tenuis*, Synechococcaceae abundance derived from the tissues correlated negatively with the only significant physiological parameter; Symbiodiniaceae density normalized to protein contents ($p = 0.025$). In the mucus of *A. millepora*, the abundance of Pirellulaceae was positively correlated with $NO_2^-/NO_3^-$ ($p = 0.035$) and negatively correlated with TSS ($p = 0.019$), while OCS155 was positively correlated to $NO_2^-/NO_3^-$ ($p = 0.015$). Proteobacteria from the mucus of *A. tenuis*, Pelagibacteraceae and Halomonadaceae, were both strongly negative correlated with chlorophyll $a$ in the seawater (Pelagibacteraceae, $p = 0.013$; Halomonadaceae, $p = 0.008$). Additionally, Halomonadaceae correlated negatively with $NH_4^+$ ($p = 0.005$; Fig. 5 and Figs. S5 and S6).

Tissue-associated Endozoicimonaceae showed a strong significant positive correlation with Symbiodiniaceae density normalized to protein content in *A. tenuis* ($p = 0.0003$). In contrast, in the tissue of *A. millepora*, Endozoicimonaceae were negatively correlated with

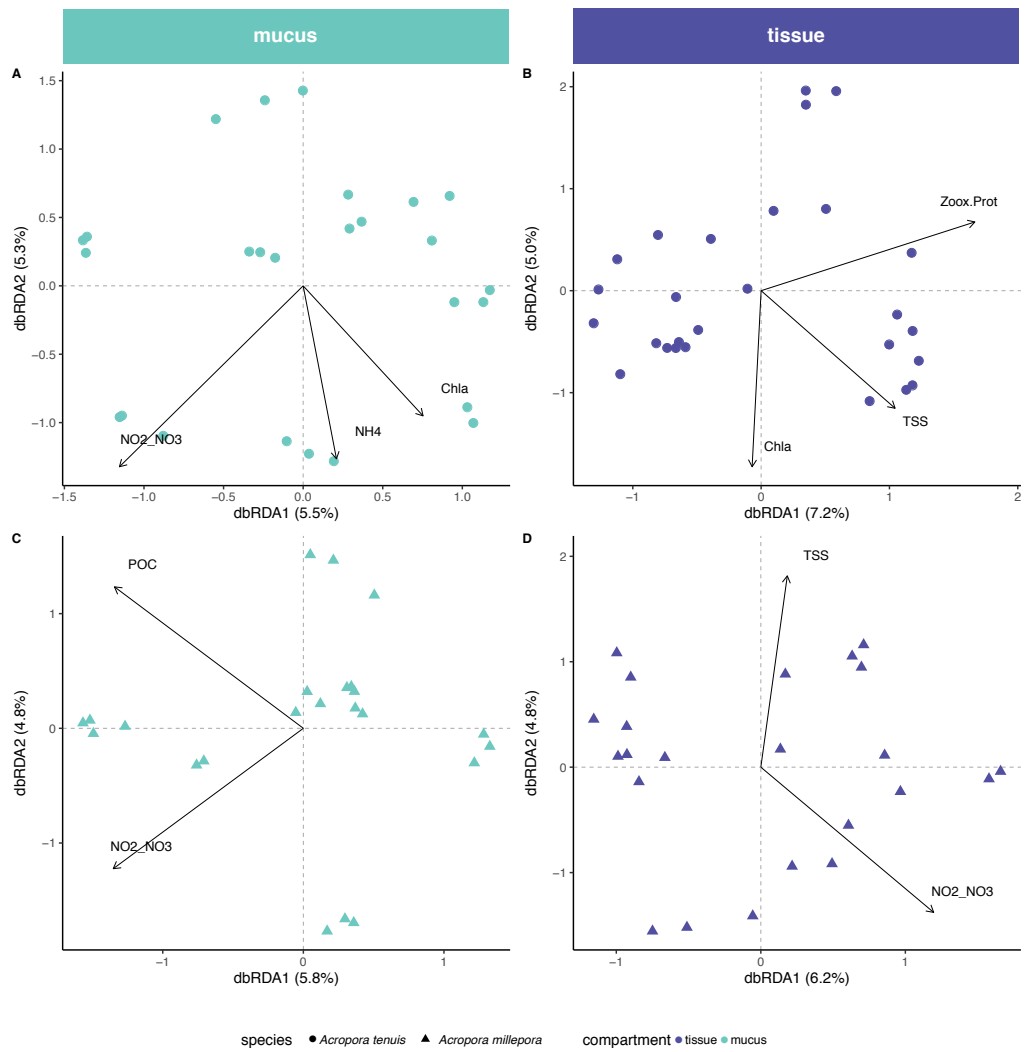

**Figure 4 Environmental and physiological drivers of the *Acropora tenuis* and *Acropora millepora* microbiomes.** Distance-based redundancy analysis (db-RDA) of the relationship between environmental/-physiological variables and the relative abundance of bacteria in (A) mucus and (B) tissue microbiome of *A. millepora*, and (C) mucus and (D) tissue microbiome of *A. tenuis*. Arrow length indicates the strength of the correlation between the variables and the samples (note only significant variables are shown). The selected variables explain a total of (A) 14.98%, (B) 16.44%, (C) 10.63% and (D) 10.97% of the observed variance, respectively. Environmental/physiological variables represented are the sum of nitrite and nitrate concentrations (NO2–NO3), particulate organic carbon (POC), total suspended solids (TSS), ammonium concentration (NH$_4$), chlorophyll *a* concentration (Chla) in seawater and Symbiodiniaceae density normalized to protein contents (Zoox.Prot) of coral tissue.

$NO_2^-/NO_3^-$ ($p = 0.020$), whereas the abundance of Cryomorphaceae family was negatively correlated with TSS ($p = 0.020$; Fig. 5, Table S5).

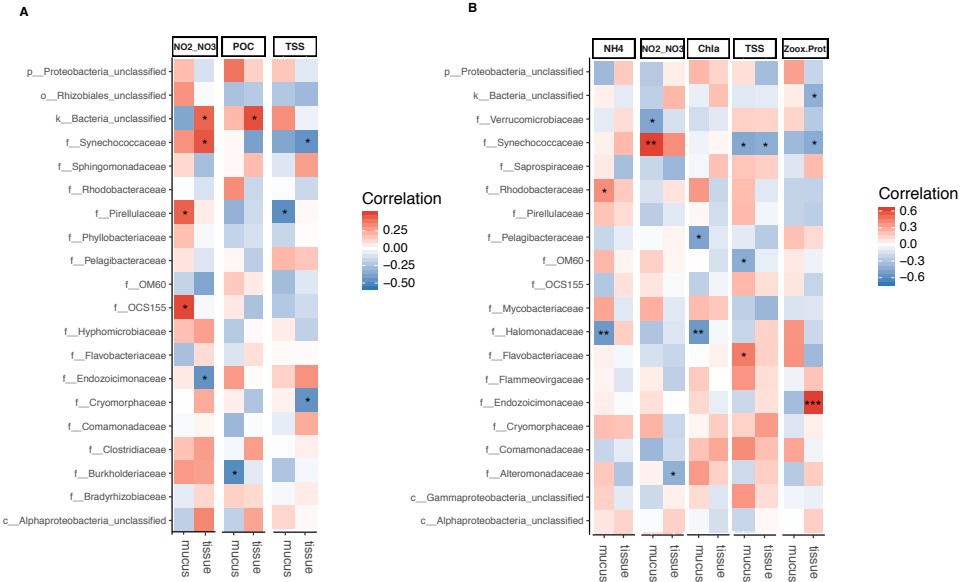

**Figure 5  Bacterial taxa significantly correlated with environmental and physiological variables.** Pearson's coefficient based correlation matrix between the 20 most abundant bacterial families and environmental/physiological variables having a significant effect on the microbiome associated to tissue and surface mucus of (A) *Acropora millepora* ($n_{\text{tissue}} = 24$, $n_{\text{mucus}} = 24$)and (B) *Acropora tenuis* ($n_{\text{tissue}} = 30$, $n_{\text{mucus}} = 28$). Significant correlations indicated by asterisks at different levels of significance (* for $p < 0.05$, ** for $p < 0.01$, *** for $p < 0.001$) after correction for multiple comparisons (using Benjamini-Hochberg correction). Environmental/physiological variables represented are the sum of nitrite and nitrate concentrations (NO2–NO3), particulate organic carbon (POC), total suspended solids (TSS), ammonium concentration (NH$_4$), chlorophyll *a* concentration (Chla) in seawater and Symbiodiniaceae density normalized to protein contents(Zoox.Prot) of coral tissue.

# DISCUSSION

Microbial communities associated with corals are continually exposed to fluctuations in the surrounding environment and the physiology of their host. Previous studies have demonstrated changes in the coral microbiome in response to thermal stress (*Ainsworth & Hoegh-Guldberg, 2009*; *Grottoli et al., 2018*; *Lee et al., 2015*; *Thurber et al., 2009*), ocean acidification (*Grottoli et al., 2018*; *Thurber et al., 2009*), organic matter enrichment (*Garren & Azam, 2012*), bleaching events (*Bourne et al., 2008*) and other environmental and physiological factors (*Glasl et al., 2019a*; *Guppy & Bythell, 2006*; *Kelly et al., 2014*; *Li et al., 2015*; *Pollock et al., 2018*). However, the coral microbiome is not homogenous across the animal and an improved understanding of the sensitivity of the microorganisms inhabiting each coral compartment is needed. This study highlights compositional differences in the bacterial communities associated with coral mucus and coral tissue, as well as with the surrounding seawater, findings that are largely consistent with previous studies (*Apprill, Weber & Santoro, 2016*; *Bourne & Munn, 2005*; *Engelen et al., 2018*; *Pollock et al., 2018*; *Sweet, Croquer & Bythell, 2011*). Furthermore, the high similarity between mucus and seawater microbiomes (see Tables S1 and S2, Figs. 2 and 3) and the high dissimilarity between tissue and seawater microbiomes suggests that the mucus microbial community
is more strongly influenced by the external environment than the tissue community. Similar results have been reported for other coral species in the Caribbean (*Orbicella faveolata*, *Diploria strigosa*, *Montastraea cavernosa*, *Porites porites* and *Porites astreoides*), where mucus and seawater shared significantly more microbial taxa than those shared by tissue and seawater microbiomes (*Apprill, Weber & Santoro, 2016*). Our results also support that mucus microbiomes are richer and more diverse than tissue microbiomes, which is a pattern corroborated by many previous studies (*Bourne & Munn, 2005*; *Koren & Rosenberg, 2006*).

Despite the host species-specificity of the coral microbiomes, some bacterial taxa were ubiquitously associated with a particular coral compartment. For example, Flavobacteriaceae and Synechococcaceae dominated the mucus of both species, while Endozoicimonaceae dominated the tissue microbiome of both *Acropora* species. However, overall microbiome composition also showed some overlap between host compartments, consistent with previous reports of overlap between the mucus and tissue microbiomes of other coral species (*Engelen et al., 2018*; *Sweet, Croquer & Bythell, 2011*). This intersection is a natural feature of the coral holobiont as both compartments are within the same host and because the constituents of the surface mucus layer are originally produced inside the tissue (*Bythell & Wild, 2011*). The sharing of some microbial taxa between compartments may also arise due to methodological challenges associated with retrieving samples that are exclusively mucus or coral tissue (*Sweet, Croquer & Bythell, 2011*), and hence these methodological limitations can obscure differences between the mucus and seawater microbiomes (*Brown & Bythell, 2005*).

## Explanatory factors of mucus microbiome variation

We hypothesized that the coral mucus microbiome, which is in direct contact with seawater, would be primarily correlated with seawater parameters, whereas the tissue microbiome would be most affected by the physiological state of the coral host. Mucus is highly hydrated: mucocyte cells release their secretions in a condensed form which then undergo a massive swelling upon hydration, forming a visco-elastic gel (*Brown & Bythell, 2005*). Surface mucus can therefore be influenced by the presence of nutrients dissolved in the surrounding seawater (*Tanaka, Ogawa & Miyajima, 2010*). As expected, environmental factors (i.e., seawater parameters) were influential in shaping the mucus microbiome of both species (*A. millepora* and *A. tenuis*), consistent with recent studies relating changes in the mucus microbiome with environmental perturbations (*Li et al., 2015*; *Pollock et al., 2018*). However, the extent of influence from environmental parameters (10% of variation) on the mucus microbiome was much lower than the influence of environment on the seawater microbiome (32% of variation), suggesting that other factors also play a role in modulating the mucus microbiome. For instance, the surrounding environment may interact with host physiology and together they alter the bacterial community structure of the mucus. Mucus is a nutrient-rich medium fueled by the photosynthetic activity of the Symbiodiniaceae (*Brown & Bythell, 2005*) and therefore it is expected that some degree of variation in its chemical composition is explained by host-Symbiodiniaceae factors. For example, *A. millepora* and *A. tenuis* at the sampling site (Geoffrey Bay at Magnetic Island)
associate with distinct Symbiodiniaceae (*LaJeunesse et al., 2018*; *Ulstrup & Van Oppen, 2003*; *Van Oppen et al., 2001*). *A. millepora* colonies were associated with *Durusdinium* (*Van Oppen et al., 2001*) whereas *A. tenuis* harbored *Cladocopium* spp. (*Ulstrup & Van Oppen, 2003*). Links between mucus chemical composition and microbiome community structure have been proposed (*Tremblay et al., 2011*). Physiological factors regulating the dynamics of production and release of the surface mucus layer could also contribute to regulating mucus microbial composition (*Glasl, Herndl & Frade, 2016*).

Fluctuations of $NH_4^+$, $NO_2^-/NO_3^-$, Chl*a* and POC in the surrounding seawater significantly correlated with the mucus microbiome variation in *Acropora* species. *Li et al. (2015)* and *Chen et al. (2011)* suggested that rainfall had a crucial effect on bacterial community variation in the coral microbiome, being mostly associated with an increase in the relative abundance of the *Bacilli* group (*Chen et al., 2011*; *Li et al., 2015*). In the present study, $NO_2^-/NO_3^-$ (and its collinear variables daylight, particulate nitrogen and grainsize of sediments; Table S3) had the greatest influence on microbiome structure, being a significant factor for both studied species. The link between rainfall and increasing nutrients (such as $NO_2^-/NO_3^-$) is well established for inshore reefs (*Fabricius, 2005*). In the current study, higher amounts of particulate and dissolved nutrients (but a decrease in TSS), corresponded to an increase in mucus-associated Synechococcaceae, Pirellulaceae, OCS155 and Rhodobacteraceae and a decrease in Halomonadaceae. For instance, Synechococcaceae in the mucus was highly positively correlated with $NO_2^-/NO_3^-$ and negatively correlated with TSS. These findings corroborate previous work in which the abundance of free-living *Synechococcus* in shallow coastal waters decreased significantly under lower nutrient (especially nitrate) and higher TSS concentrations (*Uysal & Köksalan, 2006*).

Dissolved nutrients, such as nitrogen and phosphorus, can affect coral physiology and drive changes in the associated microbial community (*Shaver et al., 2017*; *Thompson et al., 2015*). For example, organic-rich nutrients from terrestrial run-off negatively affect the health of corals and promote rapid growth of opportunistic heterotrophic bacteria (e.g., Vibrionales, Flavobacteriales and Rhodobacterales), thus affecting the overall composition of the coral microbiome (*McDevitt-Irwin et al., 2017*; *Weber et al., 2012*). In our study, the abundance of Flavobacteriaceae and Rhodobacteraceae in the mucus of *A. tenuis* correlated with TSS and $NH_4^+$, respectively. The coral holobiont, including cyanobacteria related to *Synechococcus* spp. (*Lesser et al., 2004*), can also efficiently take up inorganic nitrogen, for example, as nitrogen is required by the photosynthesis production of its Symbiodiniaceae symbionts (*Yellowlees, Rees & Leggat, 2008*). In fact, $NH_4^+$ can be assimilated by both coral and its Symbiodiniaceae (*Pernice et al., 2012*), and recent work has implicated bacteria such as *Vibrio* and *Alteromonas* in the incorporation and translocation of $NH_4^+$ into coral tissues and associated Symbiodiniaceae (*Ceh et al., 2013*). Nitrifying members of the mucus microbiome, such as ammonium oxidizing bacteria (e.g., Pirelullaceae) and archaea, are fueled by $NH_4^+$ (*Beman et al., 2007*; *Siboni et al., 2008*; *Yang et al., 2013*), and $NO_2^-/NO_3^-$ can be respired by nitrate reducers putatively active in coral microbiomes (*Siboni et al., 2008*; *Yang et al., 2013*). Interestingly, Pirellulaceae abundances in the mucus of *A. millepora* positively correlated with concentrations of environmental $NO_2^-/NO_3^-$, the products of ammonium oxidation. These nitrogen-cycling processes mediated by microbes are highly
dependent on oxygen availability, but because oxygen concentration in the mucus shows strong diel fluctuations (*Shashar, Cohen & Loya, 1993*), it is possible that both aerobic (e.g., nitrification) and anaerobic (e.g., denitrification) processes happen within the mucus layer at different times of the day. Temporal dynamics in the coral mucus microbiome are thus likely influenced by the individual and collective metabolic capabilities of the diverse assemblage of microbes and by nutrient availability in the surrounding waters.

## Explanatory factors of tissue microbiome variation

The statistical relation between the coral tissue microbiome and the environmental and physiological parameters differed between coral species. Whereas the tissue microbiome of *A. tenuis* corresponded to both environment and host physiology, *A. millepora* correlated only with environmental parameters. This difference may be associated to specific features of each species, through which *A. millepora* could modulate the internal environment and create more stable intra-tissue conditions than *A. tenuis* (e.g., via skeletal light modulation, host morphology and tissue thickness, *sensu Enriquez, Mendez & Iglesias-Prieto, 2005*). A non-mutually exclusive alternative explanation is the influence of the algal symbiont (Symbiodiniaceae) genotype associated to the host. *Little (2004)* investigated Symbiodiniaceae communities associated with *A. millepora* and *A. tenuis* on Magnetic Island demonstrating that the coral-algal endosymbiotic relationship in *Acropora* spp. is distinct between species, dynamic and flexible (corals associate with different Symbiodiniaceae types at different life stages, for example), and contributes significantly to physiological attributes of the coral holobiont. For example, different algal genotypes can affect the nutrient availability (e.g., carbon and nitrogen) in the coral holobiont (*Pernice et al., 2015*; *Bayliss et al., 2019*). Environmental factors such as seawater temperature can also lead to temporal changes in the symbiont community (*Cooper et al., 2011*; *Howells et al., 2012*; *Rocker, Willis & Bay, 2012*). As the microbiome is strongly associated to the coral holobiont, any disturbance in the host-Symbiodiniaceae relationship may have indirect effects on the microbial composition and its response to environmental and physiological factors. Other studies demonstrate the influence of Symbiodiniaceae on the host microbial community and also support the idea that these two components of the coral holobiont are finely tuned (*Glasl et al., 2017*; *Grottoli et al., 2018*; *Littman, Bourne & Willis, 2010*; *Littman, Willis & Bourne, 2009*; *Quigley et al., 2019*). In the present study, Endozoicimonaceae were strongly positively correlated with the Symbiodiniaceae density in the tissue of *A. tenuis* and negatively correlated with $NO_2^-/NO_3^-$ in *A. millepora* (see Fig. 5). These results are to some extent at odds with experimental results showing a stable dominance of Endozoicimonaceae in tissues of *Pocillopora verrucosa* irrespective of excess dissolved organic nitrogen and despite a bleaching phenomenon concomitant with structural changes in its Symbiodiniaceae community (*Pogoreutz et al., 2018*).

Besides the diversity of Symbiodiniaceae associated to each coral species, other factors can affect the coral and its response to environmental parameters, such as photochemical efficiency (Fv/Fm) and symbiont density (*Cunning & Baker, 2014*; *Da-Anoy, Cabaitan & Conaco, 2019*). For instance, *Da-Anoy, Cabaitan & Conaco (2019)* demonstrated a greater reduction of Fv/Fm in *A. tenuis* in response to elevated temperatures compared to *A.*

*millepora* and the temperature responses of the corals did not directly correlate with their associated Symbiodiniaceae. This suggests that other species-specific physiological factors could modulate the responses of the coral to the environment and, indirectly, influence the tissue-associated microbiome. One such factor is the way coral-associated microbial aggregates (CAMAs) are distributed throughout the tissue, which varies within populations and can vary among coral species (*Work & Aeby, 2014*; *Wada et al., 2019*).

Total suspended solids (TSS) was the only environmental parameter measured in the present study that significantly related to the tissue microbiome of both coral species. TSS can impact corals by limiting light availability for photosynthesis and decreasing Symbiodiniaceae densities, which can indirectly affect microbial communities (*Fabricius, 2005*; *Pollock et al., 2014*). High levels of suspended solids characterize the environment of inshore reefs such as those found around Magnetic Island. The decrease in TSS is strongly associated with an increase in the abundance of tissue-associated Synechococcaceae and Cryomorphaceae. Cryomorphaceae are typical copiotrophs in the phylum Bacteroidetes and their increase in the tissue of *A. millepora* could relate to declines in coral holobiont health.

## CONCLUSIONS

This study highlights that microbiomes inhabiting different physical microniches within the coral holobiont differ in their linkage between host and environmental factors. Microbiomes of *Acropora* spp. differed significantly among host compartments (surface mucus layer and tissue) and species (*A. tenuis* and *A. millepora*). Seawater parameters had the greatest influence on the mucus microbiome in both species whereas the tissue microbiomes showed differential patterns to environmental/host-physiological parameters, suggesting host-specific modulation of the tissue microbiome. While further research is needed to unequivocally define the drivers of coral microbiome variation, by investigating temporal variation in water quality and coral health measures and correlating these with microbial community dynamics across distinct host compartments in closely related species, this study has identified several intrinsic and extrinsic factors that contribute to microbiome composition in corals.

## ACKNOWLEDGEMENTS

We thank Michele Skuza, Neale Johnston and the AIMS water quality team for their help with analyzing the water quality samples. We thank Heidi Luter, Katarina Damjanovic and Joe Gioffre for their assistance in the field. We also thank Sara Bell, Florita Flores and Carlos Alvarez for their expertise in the laboratory. The authors acknowledge the Traditional Owners of the sea country where sampling took place. We pay our respects to their elders past, present and emerging and acknowledge their continuing spiritual connection to their sea country.

### Funding

The Australian Microbiome Initiative supported the generation of data used in this publication. The Australian Microbiome Initiative is supported by funding from Bioplatforms Australia through the Australian Government National Collaborative Research Infrastructure Strategy (NCRIS). The study was further funded by the Advance Queensland PhD Scholarship, the Great Barrier Reef Marine Park Authority Management Award, and a National Environmental Science Program (NESP) grant awarded to Bettina Glasl. Pedro R. Frade was supported by the Portuguese Science and Technology Foundation (FCT) through fellowship SFRH/BDP/110285/2015. The CCMAR team received support from FCT through UIDB/04326/2020 and SFRH/BSAB/150485/2019. The funders had no role in study design, data collection and analysis, decision to publish, or preparation of the manuscript.

### Grant Disclosures

The following grant information was disclosed by the authors:
Australian Government National Collaborative Research Infrastructure Strategy (NCRIS).
Advance Queensland PhD Scholarship.
Great Barrier Reef Marine Park Authority Management Award.
National Environmental Science Program (NESP).
Portuguese Science and Technology Foundation (FCT).
FCT: UIDB/04326/2020 and SFRH/BSAB/150485/2019.

### Competing Interests

The authors declare there are no competing interests.

### Author Contributions

- Giulia M. Marchioro, Bettina Glasl and Pedro R. Frade conceived and designed the experiments, performed the experiments, analyzed the data, prepared figures and/or tables, authored or reviewed drafts of the paper, and approved the final draft.
- Aschwin H. Engelen, Ester A. Serrão, David G. Bourne and Nicole S. Webster conceived and designed the experiments, authored or reviewed drafts of the paper, and approved the final draft.

### Field Study Permissions

The following information was supplied relating to field study approvals (i.e., approving body and any reference numbers):

Coral and seawater samples were collected under the permit G16/38348.1 issued by the Great Barrier Reef Marine Park Authority.

### DNA Deposition

The following information was supplied regarding the deposition of DNA sequences:

Demultiplexed sequences and metadata are available at the Bioplatforms Australia data portal under the Australian Microbiome Initiative (https://bioplatforms.com/projects/

australian-microbiome/). Full usage requires free registration. To search for the sequencing data, navigate to 'Processed data', select 'Amplicon is 27f519r_bacteria' and 'Environment is Marine'. To search for the Great Barrier Reef sampling sites, add two additional contextual filters, set 'Sampling Site' to 'Geoffrey Bay', and 'Sample Type' to 'Coral'.

## Data Availability
zOTU abundance table, metadata and zOTU taxonomies are available as Supplemental Files.

## Supplemental Information
Supplemental information for this article can be found online at http://dx.doi.org/10.7717/peerj.9644#supplemental-information.

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
