# Peer review of "Microbiome dynamics in the tissue and mucus of acroporid corals differ in relation to host and environmental parameters"

_PeerJ, doi:10.7717/peerj.9644_

## Round 0.1 · original submission · Major Revisions

I now have two reviews from experts in the microbiome of reef-building corals. Both have significant reservations about the current manuscript, primarily concerning three things (1) low level of replication, (2) statistical methods used on the data, (3) breadth of the conclusions, considering (1) and (2). Despite these concerns, the reviewers expressed positive comments about the insights that could be drawn from the data, if presented in a more measured manner. At this point, I would invite the authors to prepare a major revision that addresses the reviewers' comments.

Reviewer 1 ·

Basic reporting

The paper could use revision for grammar and better use of the active voice over the passive voice.
Not all sections are adequately referenced.
Further details are provided below in the general comments.

Experimental design

Some of the research questions were well defined, while others were intentionally vague, especially when drawing inferences of cause and effect which were not correct for this experimental design. Not all methods are described with sufficient detail. Further details are provided below in the general comments.

Validity of the findings

Many of the findings are presented as direct cause and effect relationships which is too far of an over-reach with the given data. Further details are provided below in the general comments.

Additional comments

This paper provides some useful but incremental information to the growing field of coral microbiome interactions. The largest weakness of the paper is that the authors swing back and forth from discussing possible correlations between microbial diversity patterns to possible external and internal variables and how external variables act as “controls,” “drivers,” “responses,” or “affects” to the microbial diversity. While testing for possible correlations between microbiome diversity and other variables are germane here, the authors must remove all language that implies any sort of cause and effect relationship, as they have no evidence for this, nor did they specifically have a proper experimental design to make such determinations.

Many of their correlations are weak and the variables chosen are so limited that the authors cannot make many solid conclusions as to how many of these variables influence the patterns in microbial diversity. In summary, the authors can certainly address their first hypothesis (microbial composition will vary based on compartment). While the relationships between the mucus layer and the overlying water column are germane (but perhaps not fully supported), the discussions concerning the tissue microbial community and “changes in the physiology of the host” are exceptionally weak and not justified without better data and a much more sophisticated controlled study.

Other issues:
The concepts of alpha and beta diversity are not mentioned in the introduction clearly, so the context of why they are germane for this paper is weak.

There is confusion in how some terms are used which makes it difficult to understand what the authors are referring to. For example, chlorophyll a was sampled from the algal symbionts in the corals but was also sampled from the water column above the corals. The authors use several terms (and abbreviations) and it was not always clear which pigment data they were referring to in the paper.

The entire paper could use some tightening and better use of the active voice over the passive voice. I have provided a few examples below.

Abstract
-Line 37: Rather than calling these “environmental” factors, could you choose a better word? Perhaps “water column parameters”?
-Line 38: can you really say these were “drivers”?
-Line 39–41: This is far too weak and requires some specificity so the reader can understand the significance (or lack thereof) of this aspect of the study.
-The last sentence of the abstract seems redundant.

Introduction:
-Line 47: Remove “eukaryotic” as this term is redundant since all dinoflagellates are eukaryotic.
-Line 51: change, “can act in defense” to “defends”
-Line 52: There are several examples in the text where the active voice should replace the passive voice. This will also help to lower the word count and to increase readability. For example, the ending sentence on this line should be simplified to: “The coral microbiome is influenced by…” Please pay attention to extensive overuse of the verb to be throughout the paper.
-Line 67: Remove “animal” as we know that the coral is an animal.
-Line 73: Can you really say this information is crucial toward understanding the symbioses or is it more general than this since not all consortia of the microbiome are necessarily “symbiotic”?
-Line 83: I believe the correct grammatical term here would be “Symbiodiniacid” since the family name is in the plural.
-Line 95: simply saying “dinoflagellate” or “algal” here sounds much better than using the family name. Again, you are using a plural followed by another plural word and this is not grammatically correct. A comma should go after “cells”.

Methods:
-More details are needed on the water and sediment samples and how they were collected. At the very least, some citations should be provided here. Simply stating that you took these samples “according to standard procedures of AIMS” with no citations is not sufficient for another person to replicate your study.
-Line 165: This section is missing an appropriate reference for the chlorophyll a extraction and quantification protocol.
-Line 178: Please specify that these are “Soluble” proteins.
Please remove the last sentence in this section, as this is redundant and assumed for the protocol.
-Line 198: Please state the number of independent hemocytometer loadings as the technical replicates here for your counts. Providing the number of squares is not as informative.
-Line: 206: I think you mean a Venn diagram was “constructed” here.

·

Basic reporting

I very much enjoyed reading this work by Marchiroro/Glasl et al. It was easy to read, had interesting visuals and the results were compelling and interesting. Overall it is a nice piece of microbiome science that adds to our knowledge about how the portions and key taxa respond to individual variables such as nutrient concentrations and/or particulate carbon. This aspect of the work makes it novel and insightful.

There are a few issues with the text and visualizations that need to be conducted to make the work acceptable for publication.

1. For the visualizations I found it strange that despite having ‘season’ being a primary variable in the experimental design, none of the visualizations have samples split by season. If the authors can prove that season was not a main effect I might understand this, but based just on the ANOVA richness data (p=0.053….think about what this really means in the context of your design) I suspect season has an interaction that limits your interpretation of some of the other factors driving changes in your community composition and individual groups of taxa changes. So please revised your visualizations (Figs. 1, 2, 3) to include from which season the samples came.

2. Further, while I always like bubble charts use of mean relative abundance in figure 1 does not show the large amount of variation of the taxa across the different sample categories. For full transparency you need to revise this to show how certain taxa are present in a wide range of abundances despite being in similar variable categories (e.g., host, season, compartment). For example, in your descriptions the Endos exhibit a very large range of variation but the reader can’t tell what the variation is being driven by (see Lines 271-273). If you use a box plot or even just show every sample in clusters on the bubble chart we’d be able to see this more completely to really access what is going on here.

3. Also Figure 2 is both underwhelming and somewhat inaccurate. You can make Venn diagrams scale so that the overlap of each category represents the relative number of similarities. Thus in your diagram there would be a very large overlap in mucus and seawater and little overlap in tissue and seawater….which I believe is the point of this figure. Right?

4. For Figure 4, I think this would be easy to follow if you removed the legend and put headers above and on the left side to indicate if the figure was tissue or mucus or from A tenuis or A millepora. Lastly my favorite part of this paper is your last figure. Again showing the variation might be nice. You can use a heat map style for this which would add more columns arranged by sample type (essentially you’d show all three of each sample category instead of the mean). To simplify it you could also only show the significantly different taxa.

Experimental design

The major flaw of this paper is its experimental design, although I don’t think that this warrants a rejection. The authors, despite the low replication for each compartment/species/time (n=3) still find important positive results that are interesting, novel, and worthy of dissemination. However, I think the authors should be wary of making strong statements concerning the negative results they find. For example, the differences between significant tests for the A. millepora and A. tenuis maybe due to lower sample sizes (lines 244-245) in one host. The low power of the experiment may also be driving the absence of differences in seasonality which is a unexpected and somewhat surprising finding that raises some red flags. To this point, the authors state in the richness section that ‘season’ was not a significant factor, but then they list a p value of 0.053. Now that’s pretty darn close, begging the question of whether this ‘negative’ result was due to the small number of samples. As you will see in the section on validity of the findings, I think the use of a GLM would help in this regard. You may see more clear main and secondary effects of difference variables, with season, I imagine, being a significant interactor. But currently it’s difficult to tell because the data are not visualized this way.

Validity of the findings

I have some questions about the microbiome analysis.
1. Why did you choose to use primers that are no longer standard? The V4-V5 (515-806) has become the most widely used and thus databases are likely to contain more taxonomic information using this region instead of the one you chose 27F-515. I think a description for the rational for this is important.
2. Line 148. Does this mean that you removed any sequences with one or more N’s or that you limited to those with more than a defined number of N’s? If the later please state the number of N’s that was permissible.
3. Did you ensure that when you generated the zOTUs/ASVs table that you trimmed to the identical sequence length to ensure that additional bases were not considered additional zOTUs? If not…you must do this. If yes please state this.
4. Why did you use an ANOVA for your diversity metrics? Most microbiome data is not normal and unless a transformation was done (none was mentioned) it is unlikely that your data will meet the assumptions of the tests. Further, in our experience GLMs perform better (aka better residuals) than an ANOVA for coral microbiome data (see Maher et al., Scientific Reports 2019 as an example). I recommend that you consider this approach in place of the ANOVAs when comparing your richness data.
5. Line 215. I did not know that these normalizations were “common.” In fact I suspect that they are not really standard. Please further defend the use of these as they are essential to your data analysis and interpretation. Also please cite additional literature other than your own paper regarding these normalizations.
6. Did you use a multiple comparisons correction for your composition data analysis?

Additional comments

I really enjoyed reviewing this paper and think that with some edits it could be a really nice contribution to our knowledge about what drives variation within and between coral microbiomes. However, I feel like the emphasis on the finding that the tissue compartment is less sensitive to the environment is not very compelling. Several papers have done this and with much better datasets to confirm it, including our own that used over 650 samples. In fact when I first read the title I kind of groaned cause I felt like this was a settled topic, and I didn’t want to review another paper on this topic. But when I read the paper I found that it was much more compelling than the title and abstract let on. Basically I think you undersell your work here. Importantly the novelty of your work is in the linkages between shifts in community members with different environmental factors that you measured. These findings have important implications in management and restoration implications and thus should be highlighted more completely.

Minor comments:

Line 91. Prokaryote? Are we still using this word? I think given the recent revisions to the tree of life, this term needs to be retired completely. Please use bacteria and archaea if that is what you mean.

Line 91. In our Pollock et al., paper we actually found that when you look across phylogenetically diverse coral hosts, the mucus is actually less diverse than tissue across the board!

Line 97-99. I would revise this hypothesis as it has been addressed several times before and is not the most compelling hypothesis you could come up with based on your introduction. This is not a requirement, it’s a stylist thing, but one I think would elevate your work.

Line 116. What size were the nubbins?

Line 121. I’m confused. Where the nubbins and sterivex snap frozen or just the sterivex?

Line 141. Again you need to address why you used these primers and not the Prada/Apprill et al., ones.

Line 206. zOTUS?

Line 248. Did you look at richness at any other level of taxonomic classification (e.g., genus?). Also did you look at phylogenetic diversity? Seems appropriate.

Line 256-257. I am a tad confused by this statement. Do you mean that species x time and water x time had no effect on the community? Or do you mean that when you look at all samples there was no effect? Seems unlikely that the water did not change across time. Again maybe using a GLM would clarify these main and interacting effects.

Lines 289-321. I really like this section and think you could expand on it to increase the impact of the work. Maybe some of the supplements could be brought to the main text and that Venn Diagram get moved to the supp.

Line 338-342. These are very cool findings! You don’t talk about it at all in the discussion! You maybe want to elaborate.

---

## Round 0.2 · Minor Revisions

Reviewer 1 has returned a review of your revision, and finds the manuscript improved, but would like to see additional minor revisions.
The reviewer poses specific questions linking the coral microbiome with coral physiology that should be addressed.

Reviewer 1 ·

Basic reporting

Overall, this paper has been improved. I still feel as though the link between the microbial community and the coral "physiology" is tenuous, and the authors could do a better job framing this. For example, why / how does the Chl a. level of the symbiotic algae relate to the bacteria? There is probably some interesting hypotheses out there if one digs into the phytoplankton literature and how other microalgae (perhaps dinoflagellates that are notoriously mixotrophic) live with some forms of bacteria. I have tried to offer other suggestions below as well. I am also not happy with some of the language that still reaches a bit too far and is assigning causation with you really need to adhere to the correlations and let them speak for themselves, and I have suggested alternative phrases.

Experimental design

no comment, please see section 4.

Validity of the findings

no comment, please see section 4 and my comments above in section 1.

Additional comments

Last sentence of the abstract is a bit vague. It would be nice if you could quickly encapsulate a few of the "novel insights" "implications" for restoration or management that you allude to here. Critically, you really do not ever tell the reader how exactly your results fit this point later in the paper either.

Line 322: remove "significantly"

Line 357: I would add "in the Caribbean" after "coral species"

Line 395: remove "have previously been shown to"

Line 397: for "were associated with Symbiodiniaceae in the genus Durusdinium," remove "Symbiodiniaceae in the genus"

Line 398: change "was associate to the genus" to "harbored" and add .sp following Cladocopium.

Line 399: Remove the sentence that begins, "These corresponded…"

Line 405: change "structured" to "correlated to"

Line 413: change "were concomitant" to "corresponded"

Line 425: add "the" between "study," and "abundances" and drop the s in abundances.

Line 449 and 450, I am still not in favor of using words like "responded" here since you did not test for a response per se. I would stick to "corresponded" or "correlated"

Line 455–460: It may be useful here to be more explicit about this relationship and perhaps the different symbiotic algae are contributing different kinds and levels of organic C and N that may also affect your microbiome here.

Line 474: This seems like a good place for a paragraph break.

Line 477: The paper by Da-Anoy and colleagues is a spurious argument, as multiple studies are now documenting physiological variability among different genotypes of the same algal species. Dig deeper and you will find examples across different strains of Breviolum minutum, Durusdinium trenchii and others. Please do not assume that having the same ITS2 sequence means you are working with only one genotype of alga, as this would be incorrect. This also weakens your point here.

Line 485: What if what you are calling total suspended solids also include particulate organic matter (POM). There is an point to be made that considerably more work is needed in comparing coral trophic dymanics to their microbial consortia here.

Line 500: change "responses" to "patterns".

Line 505: change "we begin to disentangle" to "this study has identified several intrinsic and extrinsic factors that contribute…"

---

## Round 0.3 · accepted · Accept

Thanks for the clear and careful responses to the last round of review comments. I am recommending that your revised manuscript be accepted for publication.